# Topological soliton molecule in quasi 1D charge density wave

Taehwan Im[1,2], Sun Kyu Song[1], Jae Whan Park [1] & Han Woong Yeom [1,2] ✉

Soliton molecules, bound states of two solitons, can be important for the informatics using solitons and the quest for exotic particles in a wide range of physical systems from unconventional superconductors to nuclear matter and Higgs field, but have been observed only in temporal dimension for classical wave optical systems. Here, we identify a topological soliton molecule formed spatially in an electronic system, a quasi 1D charge density wave of indium atomic wires. This system is composed of two coupled Peierls chains, which are endowed with a $Z_4$ topology and three distinct, right-chiral, left-chiral, and non-chiral, solitons. Our scanning tunneling microscopy measurements identify a bound state of right- and left-chiral solitons with distinct in-gap states and net zero phase shift. Our density functional theory calculations reveal the attractive interaction of these solitons and the hybridization of their electronic states. This result initiates the study of the interaction between solitons in electronic systems, which can provide novel manybody electronic states and extra data-handling capacity beyond the given soliton topology.

With the development of information technology, it has become crucial to process vast amount of information dispationlessly[1–3]. In this aspect, topological solitons in one dimensional electronic systems or non-topological solitons in optical fibers have obvious merits as an information carrier since they are highly mobile with immunity against scatterings[4,5]. Moreover, beyond typical $Z_2$ topological solitons with digital information, recent experimental works have identified $Z_3$[6] and $Z_4$ solitons[7] in 1D charge density waves, which deliver ternary and quarternary information, respectively, to extend substantially the data-carrying capacity of solitons. On the other hand, pairs of non-topological solitons with favored temporal separation were discovered in optical systems and discussed to multiply data carried by solitons[8–13]. That is, not only annihilating each other, the collision by another soliton can change a soliton state by forming a soliton bound state to create an extra type of data carriers. In a more general context, topological solitons are observed in various materials and field systems such as superfluids, superconductors, chiral magnets, Bose-Einstein condensates, high density quantum chromodynamics, and the Higgs doublet model, so that their bound states can lead to exotic manybody states[14].

In a simplified model, two adjacent $Z_2$ solitons in an electronic system with opposite signs of the phase shift in the same wire can have an attractive interaction[15,16]. When they become very close, their phase tails would overlap to result in a repulsive force and a proper balance of these interactions may stabilize a bound state of solitons. Thus, the formation of a stable bound state of solitons would depend largely on materials parameters and only temporal bound states of light wave (non-topological) solitons have been identified within a special type of optical fibers[8–13]. The observation of a topological soliton molecule, or a pair of interacting topological solitons has been elusive[17], to the best of our knowledge, and the interaction between solitons has rarely been accessed in experiments. A few previous scanning tunneling microscopy works for 1D charge density waves noticed local phase perturbations longer than individual solitons[18,19] but their nature as a collection of interacting solitons has not been revealed as limited by insufficient spatial resolution, the lack of proper spectroscopic measurements, and various extrinsic defects to interfere[20–24].

Here, we identify a soliton molecule appearing in a model quasi 1D charge density wave (CDW) insulator of In atomic wires on the Si(111) surface by scanning tunnelling microscopy and spectroscopy

[1]Center for Artificial Low Dimensional Electronic System, Institute for Basic Science, Pohang, Korea. [2]Department of Physics, Pohang University of Science and Technology, Pohang, Korea. ✉e-mail: yeom@postech.ac.kr

(STM/STS) measurements and DFT calculations. This system features three different types of solitons and the soliton molecule is composed of two different types of chiral solitons in one wire with, characteristically, a net zero phase shift and a hybridized in-gap state, or a molecular state of solitons.

## Results

The surface layer of one monolayer of In atoms on Si(111) is composed of an array of two zigzag In chains packed between Si chains with a $4 \times 1$ unitcell ($\times 4$ is perpendicular to the wires, see Supplementary Fig. S1a). 1D metallic bands are formed mostly along In chains, which drive a Peierls distortion together with a shear distortion below the transition temperature of 125 K[7,25,26]. This system was found to be mapped into two Peierls chains with a finite interchain interaction (see the schematic model of Supplementary Fig. S1a), which result in a $Z_4$ topology and three different types of solitonic domain walls (Supplementary Figs. S1b and c). Two of these solitons, so called right- and left-chiral solitons due to their chiral symmetry, are depicted in Fig. 1a and b. As shown in the structure models, they have different structures but a common $\pi$ phase shift on the upper In chains within a double Peierls chain. STM images (Fig. 1a and b) show that the phase shifts of right- and left-chiral solitons flip gradually the orientation of CDW unitcells in clockwise and anticlockwise directions, respectively. These solitons can be distinguished by their distinct STM images as well as by STS measurements since they have different atomic structures and localized electronic states within the CDW energy gap[7] (see Supplementary Fig. S2). Yet another kind of a soliton, the non-chiral soliton, with phase shifts on both upper and lower chains, is not relevant with the present discussion.

While a non-chiral soliton is very mobile to hinder the STM observation, chiral solitons are well trapped by pinning defects or tend to aggregate into a 2D domain wall due to the interwire interaction[7,27]. An STM image for a cluster of solitons and defects crossing three In wires are shown in Fig. 1c. It contains two extrinsic defects on the top and bottom wires, which can be distinguished clearly with bright protrusions out of the regular registry of CDW unitcells in the empty state image (see Supplementary Fig. S3)[20–24,28]. Each of these defects[28] pins a left-chiral soliton (indicated by green bars and green dashed circles)[7,27]. The combination of a defect and a chiral soliton results in a total $\pi$ phase shift (a translation by $a_0 = 0.384$ nm, the lattice constant of Si) of CDW as indicated in Fig. 1d. In stark contrast, no extrinsic defect and no net phase shift (the length of even multiples of $a_0$ between CDW maxima) is identified in the central wire while the STM image itself obviously exhibits local disturbance of CDW unitcells (Fig. 1e and supplementary Figs. S4, S5 and S6).

This type of an object is not popular but can repeatedly be observed (see Supplementary Fig. S7). Its detailed structure is shown in a high-pass filtered STM image and its phase map of Fig. 1f, g (see Supplementary Figs. S8 and S9). One can notice that the phase disturbance occurs only along the upper part of the chain and two very short $\pi$ ($a_0$) shifts (see the parts with green and red arrowheads overlapped) nullify the total phase shift (see Supplementary Fig. S9 for the quantitative measurement of the phase shift). The net zero phase shift cannot be explained by any individual solitons observed previously and its topography does not match with any known defects of the present system. The double $\pi$ phase change and the overall topographic shape (Fig. 1e and Supplementary Fig. S10) immediately lead to the microscopic structure of a right- and a left-chiral soliton put

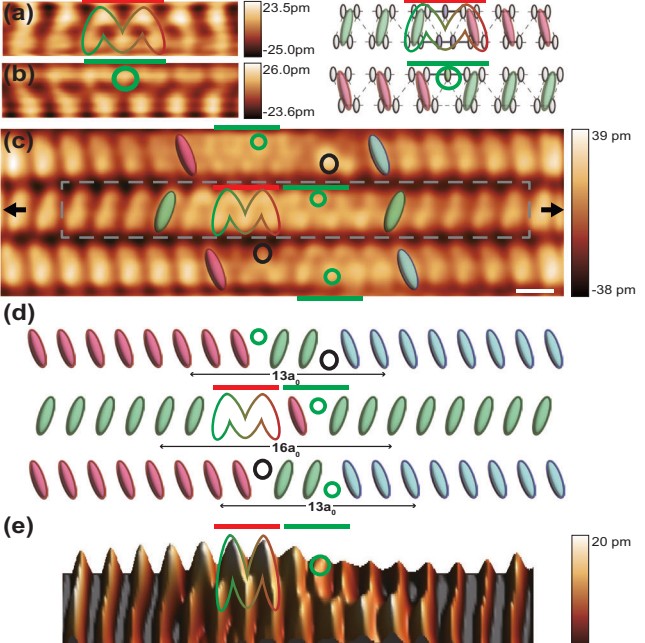

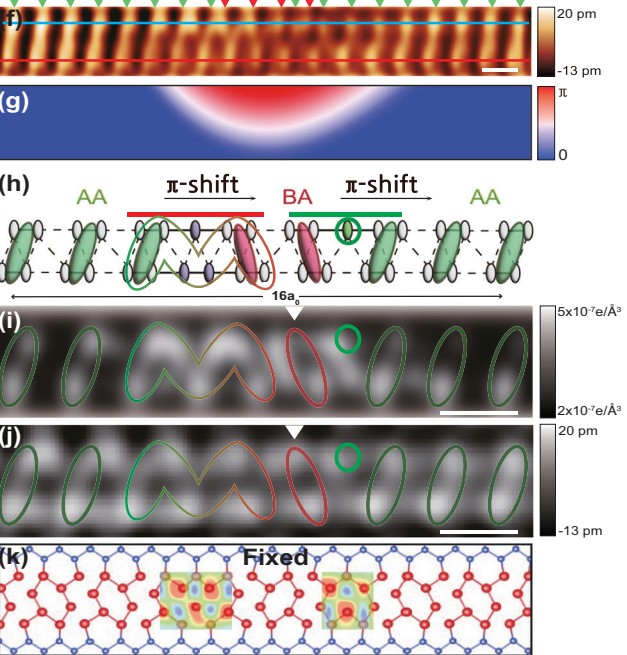

**Fig. 1 | STM topographs and atomic structures of a soliton molecule.** Filled-state STM images at T= 78K (sample bias $V_t = -500$ mV) of **a** a right- and **b** a left-chiral soliton. **c** Filled-state STM images ($V_t = -500$ mV) for a part of a 2D domain wall with solitons and defects. Defects are represented by black circles. The two black arrows represent the start and end points of the STS measurement in Fig. 2. **d** Schematics of CDW configurations across defects and solitons of **c**. **e** High-pass filtered and 3D converted image within the gray dashed square in Fig. 1c. **f** High-pass-filtered filled state STM image of the central wire in Fig. 1c. (Supplementary Fig. S8) and **g** its phase obtained by the lock-in technique[36, 37]. The green and red arrow heads represent the positions of in-phase and out-of-phase (shifted by $a_0$) CDW maxima of the wire. **h** Schematics model of a right- and a left-chiral soliton neighbored in a

double Peierls chain, which matches the local CDW variation of **a** and **b**. Ellipses indicate orientations of CDW unitcells. Grey and green circles indicate the phase-shifting sites of soltions (see Fig. 1a and b). **i** Simulated ($V_t$ = -200 mV) image with Gaussian blurring and **j** high-pass filtered experimental STM image at filled states ($V_t = -500$ mV) for a soliton molecule. The green and red ellipses represent CDW orientations and the green circle (and the 'M' shape) the phase shifting site. **k** Atomic structure model optimized in DFT calculations for a soliton molecule with a right- and a left-chiral soliton. Red and blue balls represent In and Si atoms and ellipses the CDW orientations. Calculated charges are also plotted at the phase shifting sites (the center of the solitons). Length scale bars in STM images represent 1 nm.

together very closely as shown in Fig. 1h (see Supplement Fig. S11). The total length of this object ~ $16a_0$ corresponds the length of two chiral soliton sharing two CDW unit cells in between (red ellipses in Fig. 1h) since the length of a single chiral soliton is about $9a_0$[7,29].

We perform DFT calculations to verify the structure model with two solitons coupled. Since the consideration of a huge structure with many coupled chains, a few defects and a few solitons shown in Fig. 1 is not feasible, we set up a supercell of two chains with a length of 12 CDW unitcells, one with two solitons and the other without. In this supercell, two solitons of opposite chirality have an attractive interaction (see Supplementary Figs. S12–14) with an optimal distance of $5a_0$ or $7a_0$, which is consistent with the experiment. The optimized structure is shown in Fig. 1k and its simulated STM image (total charge plot) is compared with the measured one (Fig. 1i and j). One can find reasonable agreement between the simulated and the high-pass filtered (Supplementary Fig. S8) STM images, which supports the present structure model; the only apparent discrepancy is the shift of the protrusion between the left- and right-chiral solitons (indicated by the arrow heads in Fig. 1i and j). In particular, the characteristic features of the right- and left-circular solitons can be well recognized in this comparison as depicted by a M-shape and a dashed circle. The very fine atomic structure within the molecule may need further refinement in the DFT calculation, since it does not include the defects and solitons in the neighboring chains due to the technical limitation.

The existence of two solitons in proximity, however, does not guarantee the formation of a soliton molecule. What is essential is the intersoliton interaction or the molecular level formation in electronic states. Figure 2a shows the STS data measured along the two solitons bound, the middle of the central wire in Fig. 1c [indicated two black arrows]. One can see the clear CDW gap around the Fermi level all along the wire (see dashed lines for the valance and conduction band edges). In addition, one can easily notice that the top of the valence band is pushed apparently toward the Fermi energy at the position of two solitons as well as an enhancement of the LDOS at the conduction band edge (Fig. 2b). It is apparent that the soliton pair has its own characteristic fingerprint in its electronic states in the CDW band gap region above the valence band top. It should also be noted that this in-gap LDOS feature is consistent over the two solitons while the individual right- and left-chiral solitons have clearly distinguished in-gap states near the valence and conduction band edge, respectively[7] (Supplementary Fig. 2). This indicates apparently the existence of an electronic interaction between two solitons.

In order to understand this characteristic in-gap LDOS feature, we go back to the DFT calculation for the structure shown in Fig. 1k. The DFT LDOS, shown in Fig. 2b and d, seem to capture the main spectroscopic feature of the experiments. Namely, there is a strong in-gap state above the valence band edge (green arrow head in Fig. 2) and the state at the conduction band edge (red arrow head) become slightly sharp and enhanced. Note that the current DFT calculation based on local density approximation was known to largely underestimate the band gap of the present system[30,31], which lets us insert an energy gap of 0.18 eV at the Fermi level to match with the experimental observation (see Supplementary Figs. S15–18 for the details of the band gap correction). This correction corresponds to considering extra momentum-independent exchange energy. The other limitation of the present DFT calculation is the appearance of the DOS feature at around 0.3 eV. This feature is absent in more sophisticated calculations and is not related directly with the formation of CDW and solitons[7,30,31].

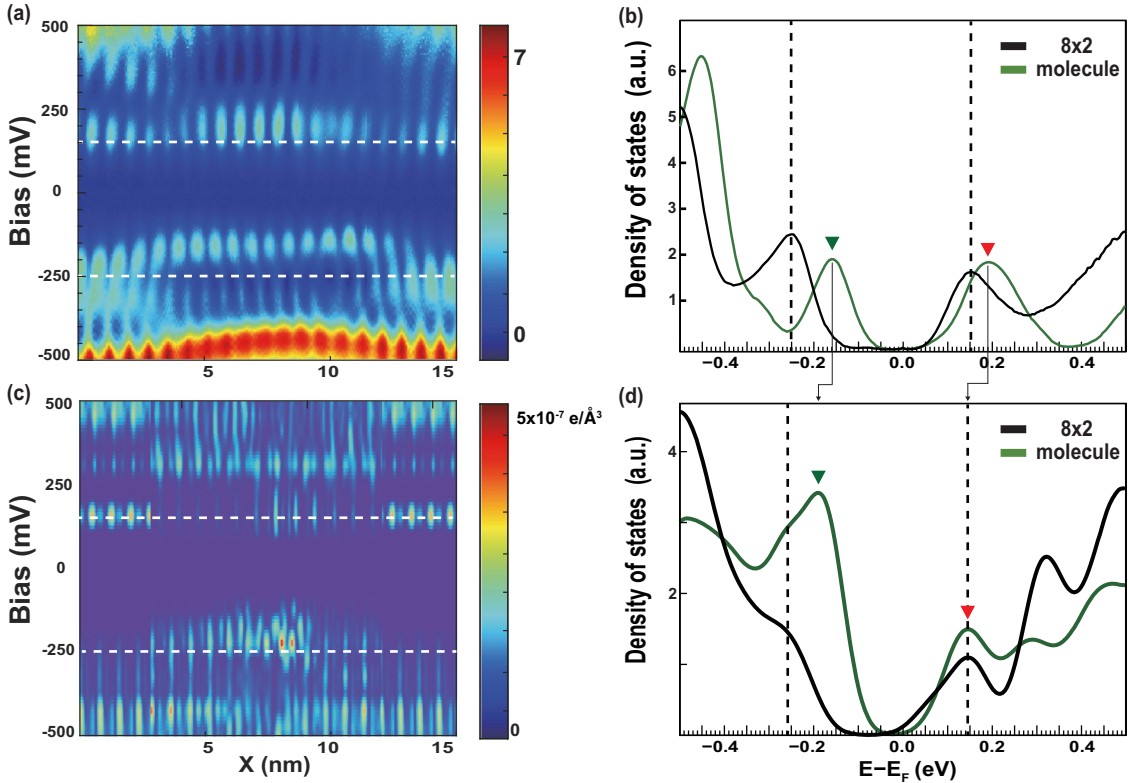

**Fig. 2 | Electronic in-gap states of a soliton molecule. a** Experimental STS line plot and **c** theoretical STS simulation crossing a soliton molecule (indicated by two black arrows in Fig. 1c). The STS simulation was obtained by integrating charge density in an interval 0.05 eV at constant height away from the top indium atom. **b** Averaged point STS spectra and **d** calculated LDOS for a pristine CDW unitcell (black) and a soliton molecule (green). Vertical dot lines represent the edge of the CDW energy gap (-0.25 - 0.15 eV). The in-gap state (the enhanced LDOS) of the soliton molecule is marked by a green (red) triangle **b**, **d**. The calculated data are broadened by 0.04 eV to account for the temperature and instrumental broadening of the experiments and added with an energy gap of 0.18 eV at the Fermi level to compensate the underestimated band gap (see Supplementary Figs. S16, S17 and S18. for detailed explanation).

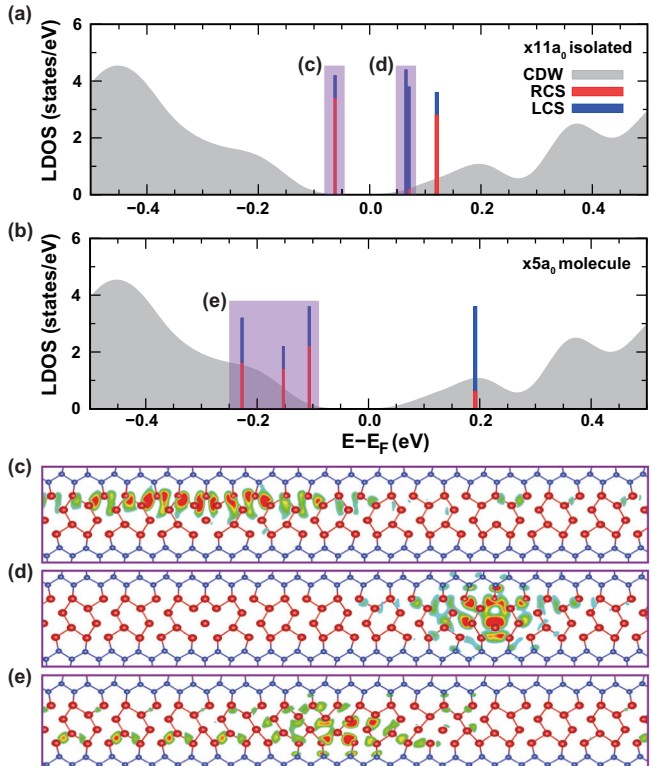

**Fig. 3 | Molecular orbital nature of in-gap states of a soliton molecule.** Calculated LDOS of **a** two isolated solitons (with a distance of $11a_0$) that do not interact with each other and **b** the soliton molecule with an intersoliton spacing of $5a_0$. For the $11a_0$ case, we fixed all indium atoms at their own CDW position to minimize the intersoliton interaction. The red and blue curves represent the LDOS of a right and a left-chiral solitons (RCS and LCS), respectively. The gray background is LDOS of the pristine CDW wire without a soliton. Vertical bars denote the energy level of eigenstates at Γ point. Calculated partial charge density of the major electronic states around the Fermi energy for the **c**, **d** isolated solitons and **e** a soliton molecule. Density of states of each energy state depicted are indicated in **a**, **b** with vertical bars.

We then trace the origin of the prominent in-gap state of the soliton molecule. Figure 3 compares the local energy levels calculated for a pair of right- and left-chiral solitons with a separation of $11a_0$ and $5a_0$, which represent a non-interacting and an interacting soliton, respectively. The uninteracting right-chiral soliton has two states near valence and conduction band edges but the uninteracting left-chiral soliton has two states degenerated at the conduction band edges. These electronic states interact strongly when the solitons are put closer to yield hybridized states at the valence band edge. The charge density plot for this in-gap hybridized state in Fig. 3e indicates the nature of a molecular level.

## Discussion

We finally note on the stability of the soliton molecule. All molecular cases observed are neighbored by solitons and defects in neighboring wires; the soliton molecules are sandwitched by defects/solitons in two neighboring wires or in proximity with defects/solitons in one of neighboring wires (Supplementary Figs. S4 and S7). Thus, it is reasonable to assume that the formation of a soliton molecule is aided by defects/solitons in neighboring wire(s) to prevent annihilation and/or thermal dissociation. In support of this, our fully relaxed DFT calculation indicates that a right- and a left-chiral soliton attract each other to collapse into a pristine CDW state and the present soliton molecule structure is a local energy minimum configuration. This configuration can be stabilized by first fixing the eight In atoms between two solitons

with all other atoms relaxed and relaxing them in the next stage (Supplementary Figs. S12–14). That is, the present soliton molecule is thought to be formed by preventing the relaxation of atoms between the solitons by most probably defects and/or solitons in neighboring wires, which are not included in the present DFT calculations. However, the intrinsic nature of the soliton molecule seems undeniable in a sense that the structure of the soliton molecule and their molecular orbitals are largely consistent over the cases observed and there is no clear sign of the structural or electronic interaction of a soliton molecule with solitons and defects in neighboring wires (Supplementary Figs. S4 and S7). While one may consider other types of soliton pairs, our previous work showed that the collision of a chiral soliton with a non-chiral soliton occurs rather frequently to merger into a soliton of an opposite chirality[32]. A pair of the same type of chiral solitons induce a net phase shift similar to a non-chiral soliton, which is highly mobile. These properties could exclude the observation of other types of soliton molecules, if any.

The present work discovers a topological soliton molecule in spatial dimension while a non-topological soliton molecule in temporal dimension was observed in optical systems. The formation of a soliton molecule leads us beyond the topological and algebraic structure of individual solitons extending the functionality of solitonic systems for informatics with low energy consumption and high data capacity. The experimental and microscopic studies of defect-soliton and soliton-soliton interactions are in its infant but has great potential for the discovery of novel manybody electronic states of diverse composite topological excitations.

## Methods

The experiment was conducted in an ultra-high vacuum system equipped with a commercial cryogenic STM system. The In/Si(111)-(4 × 1) surface was prepared by depositing one monolayer of indium atoms onto a Si(111)7 × 7 surface kept at 600 K, which was cleaned by flash heatings at 1500 K[25]. The sample was then cooled down to 78 K for STM measurements, which is well below the CDW transition temperature of 125 K[25,33]. STM images were obtained at a tunneling current of 100 pA and a sample bias of ± 500 mV. The STS data was measured using the lock-in technique with a modulation voltage of 20 mV.

### DFT calculations

DFT calculations were performed by using the Vienna ab initio simulation package[34] within the local density approximation[35]. The Si(111) surface is simulated by a periodic slab geometry with six and four Si layers for a pristine (8 × 2) structure and a (8 × 24) supercell with two solitons, respectively. In supercell calculations, we initially fix one CDW unitcell (eight indium atoms) at the center between two solitons to prevent converging to the CDW ground state. The vacuum spacing is about 13 Å and the bottom of the slab was passivated by H atoms. Electronic wave functions are expanded up to a kinetic energy cutoff of 400 eV in a plane-wave basis set and a 3 × 11 × 1 k-point mesh is used for the 8 × 2 Brillouin-zone integration. All atoms but the bottom two Si layers are relaxed until the residual force becomes smaller than 0.05 eV/Å.

## Data availability

The source data used for Figs. 1, 2, and 3 are fully available on request from the authors and are provided as in Source Data file with this paper [https://doi.org/10.6084/m9.figshare.23803551]. Source data are provided with this paper.

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

## Acknowledgements

This work was supported by the Institute for Basic Science (Grant No. IBS-R014-D1).

## Author contributions

T.I. and S.K.S. performed STM experiments and analyzed the results. J.W.P. performed DFT and calculations. T.I. and H.W.Y. wrote the manuscript. H.W.Y. conceived the project idea, made the plan, and extracted the conclusions.

## Competing interests

The authors declare no competing interests.
