## [Peer Review File · Nature Communications]

Topological soliton molecule in quasi 1D charge density waveREVIEWER COMMENTS

Reviewer #1 (Remarks to the Author):

Authors studied a structure of the quasi 1D CDW that emerges on the In wires on the Si(111) surface by using the scanning tunneling microscopy/spectroscopy technique. The authors observed soliton molecules that is a bound state of the two solitons with right and left chirality, respectively. These solitons show distinct in-gap states and net-zero-phase shift. The authors also performed a DFT calculations and claim the attractive interaction of these solitons and hybridization in the electronic structure.

An idea of the soliton molecules sounds interesting, but the authors have reported similar studies before and I don't see much novelty in the present manuscript. More importantly, there are number of places where analysis is performed incorrectly or ambiguous, and I am afraid that claims the authors make are not supported by the experimental results. The authors also claim that the experiment/simulation comparisons are in good agreement, but it looks to me that they aren't. Unfortunately, the present manuscript doesn't meet with requirements for a publication to the nature communications. Please see details below.

1. In Fig. 1, there are no color bars for the images. The authors need to add them to wherever relevant with minimum and maximum values in the manuscript. Scale bars are also missing for these images, so I can't tell the scan range of the field of view the authors studied.
2. The linecuts shown in Fig. 1(g) don't match with (f). Although I can't tell either white colors are higher or lower, peak positions in the linecuts in Fig. 1(g) don't correspond to either of the brightest or the darkest color in the image shown Fig. 1(f).
3. In p.2, right column, the authors claim that 'One can notice that the phase disturbance occurs only along the upper part of the chain and two very short $\pi(a_0)$ shifts...'. While I can see that something happens in the phase of the CDW modulation as shown in Fig. 1 (f) and (g), I am not convinced by simply looking at features in the linecut. The authors should extract a phase shift of these modulations and then carefully examine a spatial variation of

it near the boundary of the soliton molecules.

4. In Fig. 1(j) and (k), the authors compared the simulated STM image with the one obtained by the experiment and claim that “One can find the good agreement between the simulated and the observed STM images...”. However, these images look quite different and I don’t see a good correspondence at all. The simulation results show more fine structures represented by white colors, while the experimental result show rather featureless structures.

5. In Fig.2(a), the authors showed a spatial evolution of the local density of states extracted from the middle of the central wire in Fig. 1c. I notice that that there is relatively strong local density of states modulations around -250mV and -500mV, and their phase is roughly π out of phase. This indicates that a topographic signature of the CDW depends on the bias voltage used for a topography measurement since the setup current is proportional to the integrated local density states up to the setup bias voltage. Thus, I think that the STM images the authors presented in Fig. 1 would show different phases of the CDW depending on the setup bias voltages. The authors should show STM images systematically measured at different setup bias voltages, examine the CDW signatures and need to justify the data presented in Fig. 1(a), (b), (c), (d), (f), (g) and (k).

6. In Fig. 2, the authors use “High” and “Low” or use arbitrary unit without specifying actual values of the (local) density of states. Without these values, one can’t judge if the (local) density of states either gapped or not.

7. In p. 3, a paragraph starting with “In order to...”, the authors claim “The DFT LDOS, shown in Figs. 2b and 2d, seem to capture the main spectroscopic features of the experiments.”. However, I don’t see a good correspondence between them. The averaged point STS spectra in Fig. 2(b) (green curve) show two well-defined peaks, while the one obtained by the simulation show three weak peak-like features. Moreover, the average point STS spectra in Fig. 2(b) show U-shape below the CDW gap, while those obtained by the simulation show qualitatively different shapes.

8. In the supplementary Fig. 2(a), the authors show STS spectra that is represented by a ratio of the dI/dV and I/V . As I can see a gap at low sample bias voltages, I expect a zero current within the gap. Naively thinking, if the dI/dV is divided by the I/V , then $(dI/dV)/(I/V)$ should diverge within the gap. I wonder why the authors get spectra presented in the supplementary Fig. 2(a).

9. Similarly to the 2 above, in the supplementary Fig. 3, peak positions in the linecut and high/low positions in corresponding images don't match with each other. Color bars are missing as well as scale bar. It is also very strange that one obtains such a flat floor in the data that is seen in the supplementary Fig. 3(d). How exactly is the high-pass filter applied?

Reviewer #2 (Remarks to the Author):

In this manuscript the authors report combined STM and DFT investigation of topological defects in 1D indium atomic wires on Si(111) surface. By observing local electronic structure variations around these defects and comparing them with the theory, the authors claim the formation of topological soliton molecules, which might be useful in future novel information technology. While this is a very intriguing topic with interesting data presented, however, I find that part of the data is ambiguously presented and there is mismatch between the data and the theory. Therefore, I cannot recommend publication of the present manuscript, unless the following issues can be properly addressed:

1. In Fig. 1c, I find it really hard to identify by eye the two solitons in the middle line with the opposite chirality, in striking contrast to the images in Fig. 1a and 1b where these topological defects as well as their chirality can be easily seen. Even in the filtered image Fig. 1f and the linecuts Fig 1g, it is still not clear to me how to observe from the data that the upper blue line should exhibit two π phase shifts while the lower red line should exhibit no such phase shift. For example, I can see phase shifts in both blue and red curves in Fig. 1g, and the maximum amount of shift seems to be only a fraction of π . Can the authors explain this, or do the authors have better data?

2. Figure 2b and 2d are missing tick labels for the density of states (y axis in the plots). Although the authors are using arbitrary unit, it is important to indicate at least the zero value of the DOS, because whether the spectral gap is a hard gap or not can some time

affect how we interpret the data.

3. The experimental data and theory in Fig. 2b and 2d do not match well in many aspects: the overall line shape, the relative energy positions and strengths of the spectroscopic peaks, and the gap floor (rather flat in experiment but have significant variation in theory), etc. Not to mention that the overall theoretical curve has to be rescaled by almost 2 times in order to compare to the experiment (which is also pointed out by the authors in the manuscript). In this sense, I don't think the experimental features are well captured by the theory.

4. In supplementary Figure 4 the authors show the energy of structures with and without relaxation, where the relaxation is only performed to the eight indium atoms between the two solitons. Can the authors explain why they do not fully relax all the indium atoms in their calculation? Would the indium atoms in the core of the two solitons deviate from their original positions when the solitons come close? The authors state in the main text that the structure shown in Fig. 1i is optimized, but in Fig. 1i there is a label "Fixed". Can the authors clarify whether it is optimized/relaxed or fixed?

5. It is not clear where the STS line data in Fig. 2 are acquired in Fig. 1c. Can the authors make it clear, for example, draw a line in Fig. 1c to indicate the exact positions where the line is taken?

6. I am wondering how reproducible the STS features of the "soliton molecule" are? In Fig. 2 the data looks quite nice, but in supplementary Fig. 5 their spectroscopic features seem to be quite random. For example, to the left and to the right of the "molecule" the STS of the intrinsic areas do not agree.

In addition, here are some minor issues:

1. In line 11 of the first paragraph, "On the hand hand" should be "On the other hand"?
2. I am confused by the colored oval markers in Fig. 1c and Fig. 1e. It seems that there are three different colors and they are not explained in the main text or in the figure caption.
3. In the supplementary material, please change "Figure x" to something like "Supplementary Figure x" to avoid ambiguity that a reader might get during reading.
4. In line 4 of page 3, "[indicated black arrow]" is missing a "by" in front of "black".
5. In the first sentence of the last paragraph, "discovers of" should be "discovers"?

Reply to the Comments of Reviewer #1

Authors studied a structure of the quasi 1D CDW that emerges on the In wires on the Si(111) surface by using the scanning tunneling microscopy/spectroscopy technique. The authors observed soliton molecules that is a bound state of the two solitons with right and left chirality, respectively. These solitons show distinct in-gap states and net-zero-phase shift. The authors also performed a DFT calculations and claim the attractive interaction of these solitons and hybridization in the electronic structure.

An idea of the soliton molecules sounds interesting, but the authors have reported similar studies before and I don't see much novelty in the present manuscript. More importantly, there are number of places where analysis is performed incorrectly or ambiguous, and I am afraid that claims the authors make are not supported by the experimental results. The authors also claim that the experiment/simulation comparisons are in good agreement, but it looks to me that they aren't. Unfortunately, the present manuscript doesn't meet with requirements for a publication to the nature communications. Please see details below.

(Our reply) We appreciate the effort of the reviewer and his/her constructive criticism. In particular, we appreciate his/her comments on the ambiguous/incorrect part of our technical presentations in the original manuscript, which were largely improved/corrected in this revision. We apologize for a few critical errors in the previous version, which were properly corrected in this revision. We also elaborated our DFT calculations and their comparison with the experimental data. Through the proper consideration of the underestimated band gap, the agreement is significantly improved and we hope the reviewer find the consistency of our main claims and the experimental/theoretical evidence. We would like emphasize that the existence of the soliton molecule itself is proved by the STM and STS experimental data (the existence of two adjacent solitons with a net zero phase shift and the modified in-gap state) while the DFT theoretical model provides tentative atomic structure model, which is not complete at present due to the exclusion of the defects/solitons in neighboring wires.

On the other hand, we do not agree with the reviewer that the present work does not deliver novelty over the previous works including our own works. Our previous works did not reach to the formation of soliton molecules, which is distinct from the mere collection of solitons. Please note that the molecule formation has to be identified by the substantial interaction between solitons including, most importantly, the electronic molecular level formation. We believe that there has been no such report for electronic solitons in condensed matter systems so far, indicating undoubtedly the novelty of the present work.

1. In Fig. 1, there are no color bars for the images. The authors need to add them to wherever relevant with minimum and maximum values in the manuscript. Scale bars are also missing for these images, so I can't tell the scan range of the field of view the authors studied.

(Our reply) We appreciate this comment of the reviewer. We put proper scale/color bars within the corresponding figures.

2. The linecuts shown in Fig. 1(g) don't match with (f). Although I can't tell either white colors are higher or lower, peak positions in the linecuts in Fig. 1(g) don't correspond to either of the brightest or the darkest color in the image shown Fig. 1(f).

(Our reply) We appreciate this comment of the reviewer. We apologize for the inconsistent length scale in the corresponding data, which was corrected in the revised figure as shown below.

Figure caption) Figures 1 (f) and (g) in the revised manuscript. (f) Filled-state STM image containing a soliton molecule. (g) Line profiles along the blue and red lines in (f) with CDW maxima and minima guided by solid and dashed vertical lines, respectively.

3. In p.2, right column, the authors claim that ‘One can notice that the phase disturbance occurs only along the upper part of the chain and two very short $\pi(a_0)$ shifts...’. While I can see that something happens in the phase of the CDW modulation as shown in Fig. 1 (f) and (g), I am not convinced by simply looking at features in the linecut. The authors should extract a phase shift of these modulations and then carefully examine a spatial variation of it near the boundary of the soliton molecules.

(Our reply) We appreciate this important comment of the reviewer. Following the suggestion of the reviewer, we quantitatively plot the phase shift values as shown in the figure below. One can now more clearly see how the phase shift changes and can assure the characteristic double $\pi(a_0)$ shifts mentioned for soliton molecules. We add this figure into the supplements and also elaborated the corresponding part in Fig. 1.

Figure caption (a) The filled state STM image and (b) line profiles as given in Fig. 1 of the main text. (c) Plot of the distance between neighboring CDW maxima shown in (b), which quantitatively show the local CDW phase shift. The average distance between the pristine CDW maxima is about 0.8 nm. The blue line is shifted vertically to align CDW maxima with those of the red line.

4. In Fig. 1(j) and (k), the authors compared the simulated STM image with the one obtained by the experiment and claim that “One can find the good agreement between the simulated and the observed STM images...”. However, these images look quite different and I don’t see a good correspondence at all. The simulation results show more fine structures represented by white colors, while the experimental result show rather featureless structures.

(Our reply) We appreciate this comment of the reviewer. Considering this comment, we replotted carefully the simulated and measured images for the soliton molecules as shown below and in the revised manuscript. We notice that we have to use a lower bias (-0.2 eV) image in the simulation due to the underestimation of the band gap in the present DFT calculations (as detailed further below for the comment #7). The local phase shift does not change for the high and low bias images (see our reply for the comment #5 below) but the lower bias image shown below provides better agreement with the experiment in its details. Moreover, we didn’t take the experimental spatial resolution of the tip into account, which smears out fine structures in the simulated images. This point of discrepancy is significantly improved by the proper spatial broadening of the calculated features. The M shaped right-chiral soliton (dashed ovals) and trimer-like-shaped left-chiral soliton (centered at dashed circles) are in reasonable agreement between the experiment and theory. The very fine atomic structure within the molecule (as well as its energetics of a few meV scale) may need further refinement in the DFT calculation, since it does not include the defects and solitons in the neighboring chains due to the technical limitation. This point is explicitly mentioned in the revised manuscript.

Figure caption) Figures 1 (i) and (j) in the revised manuscript. (i) DFT simulated image ($V_{\text{bias}} = -0.2$ eV) of a soliton molecule with a Gaussian blurring. (j) Corresponding high-pass filtered STM image ($V_{\text{bias}} = -0.5$ eV).

5. In Fig.2(a), the authors showed a spatial evolution of the local density of states extracted from the middle of the central wire in Fig. 1c. I notice that there is relatively strong local density of states modulations around -250mV and -500mV, and their phase is roughly π out of phase. This indicates that a topographic signature of the CDW depends on the bias voltage used for a topography measurement since the setup current is proportional to the integrated local density states up to the setup bias voltage. Thus, I think that the STM images the authors presented in Fig. 1 would show different phases of the CDW depending on the setup bias voltages. The authors should show STM images systematically measured at different setup bias voltages, examine the CDW signatures and need to justify the data presented in Fig. 1(a), (b), (c), (d), (f), (g) and (k).

(Our reply) We appreciate this comment of the reviewer. In the pristine indium CDW wires, the filled and empty state topographies are out of phase across the band gap (+200 and -200 meV), hallmarking its CDW nature [Ref. 21 and 22]. This relationship is overall kept in the soliton molecule part as shown clearly below. The phase shift mentioned by the referee between the two filled states (in the LDOS map) at -250 and -450 meV is not directly related to the CDW itself but is rather incidental. The STM images shown below (also in the supplementary materials) compare topographic images and the local phase shifts of the CDW patterns at -500 (in (a) as shown in Fig. 1) and -300 meV bias. The local phase shift of the soliton molecule is consistently observed. Below the bias of -250 meV, the pristine CDW part is not clearly imaged due to its lack of density of states.

In the LDOS map below, one can analyze in more detail the phase shift behaviors in terms of the electronic states. We observe that there is no net phase shift across the molecule but the LDOS of the molecule itself has two local π phase disturbances. This is consistent with the STM topography data shown in Fig. 1. That is, the phase information in the topography and the LDOS map crossing a molecule is consistent. These data are included and discussed in the supplementary materials.

Figure caption) (a) The STS (normalized dI/dV) map shown in Fig. 2 and (b) its line plots at specified energies shown as colored arrows in (a). These line plots corresponds to the local modulation of electronic states at specific energies and those near the band gap (at $\pm 0.2-0.3$ eV, green, purple, blue, and red lines) are important for the CDW states. All these electronic states show the CDW modulations (out of phase between filled and empty states). Crossing the soliton molecule, all the modulation shows the phase shift, which but result in a net zero total phase shift. The exact location of the phase shifts along the chains (along the longitudinal direction of the soliton molecule) is slightly different between the filled and the empty states, but is consistent with the locations of the soliton molecular states in filled and empty states.

6. In Fig. 2, the authors use “High” and “Low” or use arbitrary unit without specifying actual values of the (local) density of states. Without these values, one can’t judge if the (local) density of states either gapped or not.

(Our reply) We appreciate this comment of the reviewer. Following this suggestion, we put the quantitative scales of the density of states. The LDA calculation predicts a band gap of about 0.1 eV [see the band dispersion below]. This gap wasn’t clearly shown in the previous calculations because we used a Gaussian smearing value (σ) of 0.05 eV, which is realized to be too large. See the LDOS calculations as a function of the σ value shown below. It is consistent with our experimental observation except that the theoretical band gap is much smaller than the experiment. We will discuss this quantitative difference further below (see below and the second reviewer’s comment #3). The band gap of the experimental LDOS data was made more clear in the I/V and dI/dV data shown below (with the indication of the actual values).

Figure caption) Theoretical band gap of a pristine CDW (8×2) and the soliton-molecule states. (a) Band structure of the CDW ground state in the LDA-based DFT calculation. (b) LDOS for the ground state as a function of the Gaussian smearing (σ).

Figure caption) (a) dI/dV and (b) $I-V$ curve obtained by a point STS measurement at 2.3 nm locations on a central wire containing a soliton molecule shown in Fig. 1 of the main text.

7. In p. 3, a paragraph starting with “In order to...”, the authors claim “The DFT LDOS, shown in Figs. 2b and 2d, seem to capture the main spectroscopic features of the experiments.”. However, I don’t see a good correspondence between them. The averaged point STS spectra in Fig. 2(b) (green curve) show two well-defined peaks, while the one obtained by the simulation show three weak peak-like features. Moreover, the average point STS spectra in Fig. 2(b) show U-shape below the CDW gap, while those obtained by the simulation show qualitatively different shapes.

(Our reply) We appreciate this important comment from the reviewer. We acknowledge that there was a certain degree of discrepancy and we noticed that it was partly due to the over-broadening of the calculated LDOS (we used an energy broadening of 50 meV). In particular, the lack of the U-shaped feature (the band gaps) in the simulation was due to the over-broadening as shown above.

The other, more important, problem of the current mismatch between expt. and DFT results is the great underestimation of the band gap in DFT for the current system (the pristine In/Si(111) system). This issue was well addressed previously for the present system [PRB 99, 155107 (2019)] and can partly be improved using more sophisticated functionals such as hybrid functional [PRB 102, 121408(R) (2020)] or using the GW quasiparticle calculation [PRB 99, 155107 (2019)]. Note importantly that the underestimation of the band gap does not tell that the DFT calculation cannot predict properly the ground state structure and band dispersions. As shown below with the comparison of the hybrid functional calculation, the current LDA DFT calculation, and the more accurate GW calculation, the band dispersions are consistent between these calculations but the LDA-based DFT simply underestimate the band gap size. Thus, the enlarging the energy scale, as performed in our original manuscript, should not be a good way to compensate the difference. One can more properly put extra energy gap in LDA-DFT (as shown in the third and fourth columns) for a better comparison (the red circled part around Gamma in the hybrid functional calculation is the artifact of this calculation, which deviates from the experimental results and that of the GW calculation).

Figure caption) Comparison between LDA and other functionals. (a) Heyd-Scuseria-Ernzerhof (HSE06) hybrid functional [Ref. Phys. Rev. B 102, 121408(R) (2020)]. (b) Present LDA calculation. (c) A comparison of HSE (a) and LDA (b) overlaid on each other. (d) Single-particle Green's function (G) + screened Coulomb interaction (W) band structure [Ref. Phys. Rev. B 99, 155107 (2019)].

After confirming the above, we put such an extra gap energy in the theoretical LDOS for comparison and get much better agreement between the present experimental STS and the DFT calculations (figure c below). The major spectral features of a soliton molecule, the enhanced LSOS (green arrowhead) just above the valence band edge into the band gap and the small enhancement (red arrowhead) near the conduction band edge are well captured in the calculation.

We added these discussions in the supplementary materials.

Figure caption) Comparison between experimental STS spectra (top panel) and three theoretical LDOS (bottom panel). (a) corresponds to theoretical LDOS with a gap of 0.18 eV inserted with an energy broadening of $\sigma = 0.01$ eV. (b) A Gaussian broadening of 0.04 eV was applied from that of (a).

8. In the supplementary Fig. 2(a), the authors show STS spectra that is represented by a ratio of the dI/dV and I/V . As I can see a gap at low sample bias voltages, I expect a zero current within the gap. Naively thinking, if the dI/dV is divided by the I/V , then $(dI/dV)/(I/V)$ should diverge within the gap. I wonder why the authors get spectra presented in the supplementary Fig. 2(a).

(Our reply) We appreciate this comment of the reviewer. We already provided the I/V and dI/dV data which shows the band gap clear above. We also specified in the revised manuscript that the data shown in Fig. 2(a) is the $(dI/dV)/(I/V)$. dI/dV data in general contains the tunneling probability, which is increasing as function of tunneling bias V . It is thus widely practiced in STS data presentation to use $(dI/dV)/(I/V)$, especially when one want to compare with theoretical density of states (LDOS). It is also widely practiced, in order to avoid technically the divergence mentioned by the referee, to normalize (dI/dV) by $\sqrt{(dI/dV)^2/(I/V)^2 + a^2}$ as $I \rightarrow 0$. ‘a’ represents the default value of the zero energy conductance and selects $a \ll |I/V|$ to have little effect on normalization [Phys. Rev. B 50, 4561 (1994) & *Methods of Experimental Physics*, volume 27. (1993)]. In our data, $|I/V|$ and a has the order of $\sim 10^{-10} - 10^{-9} A/V$ and $\sim 10^{-11} A/V$, respectively. This point was mentioned in the revised manuscript.

9. Similarly to the 2 above, in the supplementary Fig. 3, peak positions in the linecut and high/low positions in corresponding images don't match with each other. Color bars are missing as well as scale bar. It is also very strange that one obtains such a flat floor in the data that is seen in the supplementary Fig. 3(d). How exactly is the high-pass filter applied?

(Our reply) We appreciate this comment of the reviewer. In the prevised manuscript, we provided color/scale scale bars and corrected the line cut positions. More technical details of the high-pass filter is given in the corresponding figure caption. The flat floor part is an artificial cut-off of the data, where a defect-related strong tunneling signal appears.

Figure caption) (a) The original filled state STM images at $V_t = -500$ mV. (b) The image of the central wire after removing the slowly varying background along the wire. (c) The high-pass filtered image with fluctuating signals smaller than 10 pm in amplitude cut off from (b).

Reply to the Comments of Reviewer #2

In this manuscript the authors report combined STM and DFT investigation of topological defects in 1D indium atomic wires on Si(111) surface. By observing local electronic structure variations around these defects and comparing them with the theory, the authors claim the formation of topological soliton molecules, which might be useful in future novel information technology. While this is a very intriguing topic with interesting data presented, however, I find that part of the data is ambiguously presented and there is mismatch between the data and the theory. Therefore, I cannot recommend publication of the present manuscript, unless the following issues can be properly addressed:

(Our reply) We appreciate the positive evaluation and the constructive comments of the reviewer. We have improved the presentation of the data to eliminate ambiguity and refined our theoretical simulation to demonstrate a better agreement with the experimental data. We also checked the reproducibility further of our experimental data with some extra measurements. While there still remains a marginal quantitative discrepancy, we believe that the present calculation provides agreeable atomic and electronic structure model of the soliton molecule identified experimentally. We would like emphasize that the existence of the soliton molecule itself is proved by the STM and STS experimental data (the existence of two adjacent solitons with a net zero phase shift and the modified in-gap state) while the DFT theoretical model provides tentative atomic structure model, which is not complete due to the exclusion of the defects/solitons in neighboring wires.

1. In Fig. 1c, I find it really hard to identify by eye the two solitons in the middle line with the opposite chirality, in striking contrast to the images in Fig. 1a and 1b where these topological defects as well as their chirality can be easily seen. Even in the filtered image Fig. 1f and the linecuts Fig 1g, it is still not clear to me how to observe from the data that the upper blue line should exhibit two π phase shifts while the lower red line should exhibit no such phase shift. For example, I can see phase shifts in both blue and red curves in Fig. 1g, and the maximum amount of shift seems to be only a fraction of π . Can the authors explain this, or do the authors have better data?

(Our reply) We appreciate this comment of the reviewer. Considering this comment, we replotted carefully the simulated and measured images for the soliton molecules as shown below and in the revised manuscript. We notice that we have to use a lower bias (-0.2 eV) image in the simulation due to the underestimation of the band gap in the present DFT calculations (as detailed further below). The local phase shift does not change for the high and low bias images but the lower bias image shown below provides better agreement with the experiment in its details. Moreover, we didn't take the experimental spatial resolution of the tip into account, which smears out fine structures in the simulated images. This point of discrepancy is significantly improved by the proper spatial broadening of the calculated features. The M shaped right-chiral soliton (dashed ovals) and trimer-like-shaped left-chiral soliton (centered at dashed circles) are in reasonable agreement between the experiment and theory.

Figure caption) Figures 1 (i) and (j) in the revised manuscript. (i) DFT simulated image ($V_{\text{bias}} = -0.2 \text{ eV}$) of a soliton molecule with a Gaussian blurring. (j) Corresponding high-pass filtered STM image ($V_{\text{bias}} = -0.5 \text{ eV}$).

We also showed the phase shift involved in the soliton molecule as given below. One can now very clearly see there exist two π phase shifts on the upper part of the In atomic chain, which is separated spatially. This cannot be explained by any solitons but can be explained by our own soliton molecule model.

Figure caption) (a) The filled state STM image and (b) line profiles as given in Fig. 1 of the main text. (c) Plot of the distance between neighboring CDW maxima shown in (b), which quantitatively show the local CDW phase shift. The average distance between the pristine CDW maxima is about 0.8 nm . The blue line is shifted vertically to align CDW maxima with those of the red line.

Furthermore, in the following figure, we also put 3D renderings of the measured and simulated topography of the soliton molecules, which may help to see the existence of an unusual local structure.

Figure caption) 3D topography of soliton molecule. (a) Experiment. (b) Charge density plot integrated over the energy range of -0.2 to 0 eV in the DFT calculation. The contour interval is set to $3 \times 10^{-8} \text{ e}/\text{\AA}^3$

These data are included in the revised figure and in the supplements.

2. Figure 2b and 2d are missing tick labels for the density of states (y axis in the plots). Although the authors are using arbitrary unit, it is important to indicate at least the zero value of the DOS, because whether the spectral gap is a hard gap or not can some time affect how we interpret the data.

(Our reply) We appreciate this comment of the reviewer. In the revised manuscript, we provided proper tick labels, scale, and zero positions. The CDW gap is a hard gap in experiment as consistently reported by a few previous STM and ARPES works. The case of the DFT calculation is also a hard gap (a in the figure below) but was too much broadened in our original figure. See our reply to the comment #6 of the reviewer 1 and the figure below. With the properly reduced energy broadening of the DFT data (b in the figure below), one can clearly see the existence of the hard band gap.

Figure caption) Theoretical band gap of a pristine CDW (8×2) and the soliton-molecule states. (a) Band structure of the CDW ground state in the LDA-based DFT calculation. (b) LDOS for the ground state as a function of the Gaussian smearing (σ).

Figure caption) (a) dI/dV and (b) I-V curve obtained by a point STS measurement at 2.3 nm locations on a central wire containing a soliton molecule shown in Fig. 1 of the main text.

3. The experimental data and theory in Fig. 2b and 2d do not match well in many aspects: the overall line shape, the relative energy positions and strengths of the spectroscopic peaks, and the gap floor (rather flat in experiment but have significant variation in theory), etc. Not to mention that the overall theoretical curve has to be rescaled by almost 2 times in order to compare to the experiment (which is also pointed out by the authors in the manuscript). In this sense, I don't think the experimental features are well captured by the theory.

(Our reply) We appreciate this important comment of the reviewer. In the original submission, we overbroadened the simulated data (50 meV, which were included to explain the thermal and instrumental energy broadening of the experiments) and there is an issue of the great underestimation of the band gap in the current LDA-based DFT calculation. With a proper correction of these two effects as described in detail above for the comment 7 of the other reviewer, one can get reasonable comparison of the STS and DFT results. The major spectral features of a soliton molecule, the enhanced LSOS just above the valence band edge into the band gap and the small enhancement near the conduction band edge are well captured in the calculation.

Figure caption) Figure 2(b) and (d) in the manuscript. (b) Averaged point STS spectra (d) Calculated LDOS for a pristine CDW unitcell (black) and a soliton molecule (green).

The corresponding discussion was included in the supplements and the corresponding figures are revised.

4. In supplementary Figure 4 the authors show the energy of structures with and without relaxation, where the relaxation is only performed to the eight indium atoms between the two solitons. Can the authors explain why they do not fully relax all the indium atoms in their calculation? Would the indium atoms in the core of the two solitons deviate from their original positions when the solitons come close? The authors state in the main text that the structure shown in Fig. 1i is optimized, but in Fig. 1i there is a label “Fixed”. Can the authors clarify whether it is optimized/relaxed or fixed?

(Our reply) We appreciate this comment of the reviewer. We are regretful for the insufficient figure caption. In supplementary Figure 4, we first relaxed all indium atoms except the eight indium atoms in the middle of two solitons (see the grey regions in the figure below) and, then, we fully relaxed all indium atoms from the final structure of the first stage including those eight indium atoms. The specification “fixed” means that they are fixed in the first stage of the relaxation process. Relaxing all atoms from the beginning makes the two solitons to merge and annihilate each other into the pristine structure. As discussed in the manuscript, this, the needs of two stage relaxation, is attributed to the influence of the pinning defects in the neighboring chains, which could not be included in the present calculation. Following the reviewer’s comment, we provide all partially and fully relaxed structures in the Supplementary Materials and specified this procedure more explicitly.

Figure caption) Atomic structure of soliton molecules. (a)-(b) $5a_0$ distance. (d)-(f) $11a_0$ distance. (a), (d) Initial structure. (b), (e) The eight In atoms in the middle of the two solitons are fixed. (c), (f) Full-relaxed from the (b) and (e) structure, respectively.

5. It is not clear where the STS line data in Fig. 2 are acquired in Fig. 1c. Can the authors make it clear, for example, draw a line in Fig. 1c to indicate the exact positions where the line is taken?

(Our reply) We appreciate this comment of the reviewer. In the revised manuscript, we specified the line position for the STS line data of Fig. 2.

6. I am wondering how reproducible the STS features of the “soliton molecule” are? In Fig. 2 the data looks quite nice, but in supplementary Fig. 5 their spectroscopic features seem to be quite random. For example, to the left and to the right of the “molecule” the STS of the intrinsic areas do not agree.

(Our reply) We appreciate this critical comment of the reviewer. We note that the inconsistency of the left and the right part of the molecule seems to occur due to the different defect structures around the molecules in the neighboring wires. Note, however, that the topographic signature (the shape and the double π phase shift) of all these are the same (as shown in the corresponding figure) and they share the most important spectral feature of the soliton molecule, the in-gap state above the valence band maximum. In order to support the consistency and the reproducibility of the STS data, we performed additional experiments and show two extra molecular cases with well defined spectral data as shown below. These data replaces some of the previous data in the supplements.

In addition, here are some minor issues:

(Our reply) We appreciate all these this comment of the reviewer on the presentation and writing.

1. In line 11 of the first paragraph, “On the hand hand” should be “On the other hand”?

(Our reply) We corrected the type error.

2. I am confused by the colored oval markers in Fig. 1c and Fig. 1e. It seems that there are three different colors and they are not explained in the main text or in the figure caption.

(Our reply) We put proper explanation of the symbols in the figure caption.

3. In the supplementary material, please change “Figure x” to something like “Supplementary Figure x” to avoid ambiguity that a reader might get during reading.

(Our reply) We change the figure naming for the supplementary figures.

4. In line 4 of page 3, “[indicated black arrow]” is missing a “by” in front of “black”.

(Our reply) We corrected the type error.

5. In the first sentence of the last paragraph, “discovers of” should be “discovers”?

(Our reply) We corrected the type error.

REVIEWER COMMENTS

Reviewer #1 (Remarks to the Author):

While I appreciate the author's answers in response to my questions, and while the authors improved the analysis, I still don't think that the claims the authors made in the manuscript are well supported by the results and analysis. For example, the soliton is a well-defined physical entity as a change of the phase in the modulation, thus the phase shift is a key degree of freedom. However, an evaluation of the phase shift is still lacking in the manuscript. I don't think the present manuscript meets with the requirements for a publication in Nature Communications. Comments are as follows.

My new comments response: green

My original comments: blue

The authors responses: black

3. In p.2, right column, the authors claim that 'One can notice that the phase disturbance occurs only along the upper part of the chain and two very short $\pi(a_0)$ shifts...'. While I can see that something happens in the phase of the CDW modulation as shown in Fig. 1 (f) and (g), I am not convinced by simply looking at features in the linecut. The authors should extract a phase shift of these modulations and then carefully examine a spatial variation of it near the boundary of the soliton molecules.

(Our reply) We appreciate this important comment of the reviewer. Following the suggestion of the reviewer, we quantitatively plot the phase shift values as shown in the figure below. One can now more clearly see how the phase shift changes and can assure the characteristic double $\pi(a_0)$ shifts mentioned for soliton molecules. We add this figure into the supplements and also elaborated the corresponding part in Fig. 1.

While the figure (c) above reflects a property of the local phase shifts, it is not exactly the phase shift. I think the authors should do a better job to prove the presence of the solitons. What I suggest to the authors is to look at the phase shift of the CDW modulations as a function of position, in which the conventional CDW soliton is realized as a phase jump by 2π . To do that the authors would need to perform the "2D Lock-in" technique that has been commonly used to extract the amplitude, $A_Q(r)$, and phase shift, $\phi_Q(r)$, of a modulation at a wavevector Q , which characterizes the CDW modulation observed in the STM data (for examples, see J. A. Slezak et al., PNAS 105 (9) 3203-3208 (2008) and A. Mesaros, et al. Science 333, 426 (2011)). I suggest the authors to evaluate $\phi_Q(r)$ and to demonstrate that $\phi_Q(r)$ shows the phase jumps wherever you have solitons.

4. In Fig. 1(j) and (k), the authors compared the simulated STM image with the one obtained by the experiment and claim that "One can find the good agreement between the simulated and the observed STM images...". However, these images look quite different and I don't see a good correspondence at all. The simulation results show more fine structures represented by white colors, while the experimental result show rather featureless structures.

(Our reply) We appreciate this comment of the reviewer. Considering this comment, we replotted carefully the simulated and measured images for the soliton molecules as shown below and in the revised manuscript. We notice that we have to use a lower bias (-0.2 eV) image in the simulation due to the underestimation of the band gap in the present DFT calculations (as detailed further below for the comment #7). The local phase shift does not change for the high and low bias images (see our reply for the comment #5 below) but the lower bias image shown below provides better agreement with the experiment in its details. Moreover, we didn't take the experimental spatial resolution of the tip into account, which smears out fine structures in the simulated images. This point of discrepancy is significantly improved by the proper spatial broadening of the calculated features. The M shaped right chiral soliton (dashed ovals) and trimer-like-shaped left-chiral soliton (centered at dashed circles) are in reasonable agreement between the experiment and theory. The very fine atomic structure within the molecule (as well as its energetics of a few meV scale) may need further refinement in the DFT calculation, since it does not include the defects and solitons in the neighboring chains due to the technical limitation. This point is explicitly mentioned in the revised manuscript.

It is still difficult for me to agree with the author's claim. The guide to the eye apparently emphasizes a good correspondence between experiment and simulation results, but there are features that should be highlighted if I follow the author's standard, as highlighted in yellow above.

I can also look at this image in an alternative way. There might be a local disorder such that a local phase shift is present in the modulation, but it's so small. If I draw the circles as highlighted in yellow, no obvious phase jump in the CDW modulation is present in the first place, while I can tell there are phase jumps in the simulation.

As I described in the comment in the previous question, the authors should show and evaluate the phase shift $\phi_Q(\mathbf{r})$ of the CDW modulation in this image and demonstrate that there are phase jumps that is consistent with the calculation.

7. In p. 3, a paragraph starting with "In order to...", the authors claim "The DFT LDOS, shown in Figs. 2b and 2d, seem to capture the main spectroscopic features of the experiments.". However, I don't see a good correspondence between them. The averaged point STS spectra in Fig. 2(b) (green curve) show two

well-defined peaks, while the one obtained by the simulation show three weak peak-like features. Moreover, the average point STS spectra in Fig. 2(b) show U-shape below the CDW gap, while those obtained by the simulation show qualitatively different shapes.

(Our reply) We appreciate this important comment from the reviewer. We acknowledge that there was a certain degree of discrepancy and we noticed that it was partly due to the over-broadening of the calculated LDOS (we used an energy broadening of 50 meV). In particular, the lack of the U-shaped feature (the band gaps) in the simulation was due to the over-broadening as shown above. The other, more important, problem of the current mismatch between expt. and DFT results is the great underestimation of the band gap in DFT for the current system (the pristine In/Si(111) system). This issue was well addressed previously for the present system [PRB 99, 155107 (2019)] and can partly be improved using more sophisticated functionals such as hybrid functional [PRB 102, 121408(R) (2020)] or using the GW quasiparticle calculation [PRB 99, 155107 (2019)]. Note importantly that the underestimation of the band gap does not tell that the DFT calculation cannot predict properly the ground state structure and band dispersions. As shown below with the comparison of the hybrid functional calculation, the current LDA DFT calculation, and the more accurate GW calculation, the band dispersions are consistent between these calculations but the LDA-based DFT simply underestimate the band gap size. Thus, the enlarging the energy scale, as performed in our original manuscript, should not be a good way to compensate the difference. One can more properly put extra energy gap in LDA DFT (as shown in the third and fourth columns) for a better comparison (the red circled part around Gamma in the hybrid functional calculation is the artifact of this calculation, which deviates from the experimental results and that of the GW calculation). Figure caption) Comparison between LDA and other functionals. (a) Heyd-Scuseria-Ernzerhof (HSE06) hybrid functional [Ref. Phys. Rev. B 102, 121408(R) (2020)]. (b) Present LDA calculation. (c) A comparison of HSE (a) and LDA (b) overlaid on each other. (d) Single-particle Green's function (G) + screened Coulomb interaction (W) band structure [Ref. Phys. Rev. B 99, 155107 (2019)]. After confirming the above, we put such an extra gap energy in the theoretical LDOS for comparison and get much better agreement between the present experimental STS and the DFT calculations (figure c below). The major spectral features of a soliton molecule, the enhanced LSOS (green arrowhead) just above the valence band edge into the band gap and the small enhancement (red arrowhead) near the conduction band edge are well captured in the calculation. We added these discussions in the supplementary materials.

The authors used $s=0.04\text{eV}$ for broadening the LDOS spectra obtained by the calculation, as the STM measurement was performed at 78K. However, if I use $4k_bT$ as a thermal broadening is about 0.027eV at 78K. So, I think that the spectra are still over broadened. Any reason to choose 0.04 eV .

9. Similarly to the 2 above, in the supplementary Fig. 3, peak positions in the linecut and high/low positions in corresponding images don't match with each other. Color bars are missing as well as scale bar. It is also very strange that one obtains such a flat floor in the data that is seen in the supplementary Fig. 3(d). How exactly is the high-pass filter applied?

(Our reply) We appreciate this comment of the reviewer. In the revised manuscript, we provided color/scale scale bars and corrected the line cut positions. More technical details of the high-pass filter is given in the corresponding figure caption. The flat floor part is an artificial cut-off of the data, where a defect-related strong tunneling signal appears.

Figure caption) (a) The original filled state STM images at $V_t = -500$ mV. (b) The image of the central wire after removing the slowly varying background along the wire. (c) The high-pass filtered image with fluctuating signals smaller than 10 pm in amplitude cut off from (b).

I don't understand the procedure the authors described for the high-pass filter for the analysis. The high-pass filter is an image process that removes the slowly oscillating component so that those oscillating above the cut-off frequency (in the present case cut-off wavevector q) remain in the image. So, the figure (b) above should be the high-pass filtered image.

I don't understand what the image (c) is. Is it the same image as (b), but just changed the color depth from -24.0 pm to -13.1 pm for the minimum? More importantly, there is a $\sim\pi$ phase shift in the modulations between (b) and (c), especially in the field of view at right. I don't understand what the authors have done for this process.

Reviewer #2 (Remarks to the Author):

I appreciate Im et al. for their detailed additional data analysis and better data presentation. Although the manuscript has been improved, I find new issues arising from the authors' response, which need to be clarified before I can recommend publication of this manuscript.

1. The revised Fig. 1i and j still do not match with each other in some details. To list one such difference, I can hardly see the AA chirality feature from the left region of the experimental data, while the AA feature is very clear in the simulated image. I am afraid that the theoretical model proposed by the authors cannot be applied to the experiment. By the way, the STM topographic images are missing scale bars.

2. It does not sound a common practice to interpret DFT result by randomly shifting the energies of the filled and empty states in the opposite directions (in order to make the experiment and theory look similar). I also do not agree that the authors' LDA result is in good agreement with the GW result even after energy shifting (shown in the response figure to last-round referee 1's comment 7, as a support for shifting the energy of the bands around). Therefore, it adds further doubts on the agreement between the authors' theory and experiment.

One old issue: It seems that I still cannot find the exact locations where the Fig. 2 STS data were acquired. The black arrow in Fig. 1c points out the line where the data were taken, but the positions along that line seem to be not mentioned (or am I missing this information from the manuscript?).

Point-by-point response to the comments of the reviewers:

Reviewer #1 (Remarks to the Author):

Reviewer comment) While I appreciate the author's answers in response to my questions, and while the authors improved the analysis, I still don't think that the claims the authors made in the manuscript are well supported by the results and analysis. For example, the soliton is a well-defined physical entity as a change of the phase in the modulation, thus the phase shift is a key degree of freedom. However, an evaluation of the phase shift is still lacking in the manuscript. I don't think the present manuscript meets with the requirements for a publication in Nature Communications. Comments are as follows.

Our reply) We appreciate the constructive feedback from the reviewer. The remaining issues, we believe, are technical issues, as we made unambiguous below.

Reviewer comment 1) While the figure (c) above reflects a property of the local phase shifts, it is not exactly the phase shift. I think the authors should do a better job to prove the presence of the solitons. What I suggest to the authors is to look at the phase shift of the CDW modulations as a function of position, in which the conventional CDW soliton is realized as a phase jump by 2π . To do that the authors would need to perform the "2D Lock-in" technique that has been commonly used to extract the amplitude, $AQ(r)$, and phase shift, $fQ(r)$, of a modulation at a wavevector Q , which characterizes the CDW modulation observed in the STM data (for examples, see J. A. Slezak et al., PNAS 105 (9) 3203-3208 (2008) and A. Mesaros, et al. Science 333, 426 (2011)). I suggest the authors to evaluate $fQ(r)$ and to demonstrate that $fQ(r)$ shows the phase jumps wherever you have solitons.

Our reply) The reviewer acknowledged the phase shift itself ("reflects a property of the local phase shift") but suggested a different way to evaluate it. We appreciate this suggestion to reconfirm the local phase shift but, in principle, this is a matter of technical choice. Following the suggestion, we performed the lock-in extraction of the phase shift for five different images of soliton molecules. The result clearly shows the existence of π and $-\pi$ phase shifts on different sides of the molecule to make a total phase shift zero (2π). As we discussed in the manuscript, these phase shifts occur only on one of double In chains of a wire. This is the characteristics of chiral solitons, which constitute the present soliton molecule. We added this figure into the supplements (Supplementary Fig. S9).

Figure caption) (a) Filled-state STM image of a soliton molecule (shown in Fig. 1 of the manuscript) and (b) its phase map as obtained by the lock-in technique [J.A. Slezak et al., PNAS 105(9) 3203-3208(2008); A. Mesaros, et al. Science 333, 426(2011)]. (c) Line cuts of the phase map along the upper and lower In chains shown with matched colors in (b). The double π phase shift exists only on one In chain among two within a wire manifesting the existence of two chiral solitons in close proximity. (d) The same sets of data for four other soliton-molecule cases.

Reviewer comment 2) It is still difficult for me to agree with the author's claim. The guide to the eye apparently emphasizes a good correspondence between experiment and simulation results, but there are features that should be highlighted if I follow the author's standard, as highlighted in yellow above. I can also look at this image in an alternative way. There might be a local disorder such that a local phase shift is present in the modulation, but it's so small. If I draw the circles as highlighted in yellow, no obvious phase jump in the CDW modulation is present in the first place, while I can tell there are phase jumps in the simulation. As I described in the comment in the previous question, the authors should show and evaluate the phase shift $fQ(r)$ of the CDW modulation in this image and demonstrate that there are phase jumps that is consistent with the calculation.

Our reply) We appreciate this comment. We understand that the reviewer basically questions the existence of the systematic phase shift ($\pi + \pi = 2\pi$) of the soliton molecule we are discussing. This question is, we believe, fully answered above. This does not depend on how one groups the local protrusions.

As for the mismatch of the simulated and the experimental topography in the minor protrusions mentioned by the reviewer, we believe this is due to the different contrast ranges of the simulated and the experimental images. That is, the yellow-circled small protrusions indicated by the reviewer can also be found in the simulation as shown in the figure below (which was already provided in the

previous revision).

Reviewer comment 3) The authors used $s=0.04\text{eV}$ for broadening the LDOS spectra obtained by the calculation, as the STM measurement was performed at 78K. However, if I use 4kbT as a thermal broadening is about 0.027eV at 78K. So, I think that the spectra are still over broadened. Any reason to choose 0.04 eV .

Our reply) We appreciate this comment. There is no strong reason to use 0.4 eV , which is a representative value for a range of broadening values compatible with the experimental data. We agree that this is bigger than 4kbT value indicated by the reviewer but note that this smearing parameter represents only the electronic temperature without the lattice temperature and, thus, should not be directly compared with the real temperature of crystals in principle [Sci. Rep. 5, 16646 (2015)]. It gives only relative temperature scale and there must be other broadening factors.

Reviewer comment 4) I don't understand the procedure the authors described for the high-pass filter for the analysis. The highpass filter is an image process that removes the slowly oscillating component so that those oscillating above the cut-off frequency (in the present case cut-off wavevector q) remain in the image. So, the figure (b) above should be the high-pass filtered image. I don't understand what the image (c) is. Is it the same image as (b), but just changed the color depth from -24.0pm to -13.1pm for the minimum? More importantly, there is a $\sim\pi$ phase shift in the modulations between (b) and (c), especially in the field of view at right. I don't understand what the authors have done for this process.

Our reply) We appreciate this comment. As we already described in our previous revision, what we did is to cut off modulations less than a certain height (10 pm). That is, this method is a simple constant height-cutoff to increase the relative contrast of the CDW maxima. In fact, the same image can be obtained by removing the low frequency part of the FFT image (remove the near-zero-momentum component as shown below). We explain this procedure more explicitly in the revised supplements (Supplementary Fig. S8).

Figure caption) (a) The original filled state STM images at $V_t = -500$ mV. (b) An image of the central wire that is flattened by removing a slowly varying background and (c) its high-pass filtered image with cut-off a Γ -point from the 2D FFT of (b) and added 3 pm offset to match the maximum height of (b). (d) 2D FFT image of (b) and (e) its filtered image.

Reviewer #2 (Remarks to the Author):

Reviewer comment) I appreciate Im et al. for their detailed additional data analysis and better data presentation. Although the manuscript has been improved, I find new issues arising from the authors' response, which need to be clarified before I can recommend publication of this manuscript.

Our reply) We appreciate the positive evaluation of our revision.

Reviewer comment 1) The revised Fig. 1i and j still do not match with each other in some details. To list one such difference, I can hardly see the AA chirality feature from the left region of the experimental data, while the AA feature is very clear in the simulated image. I am afraid that the theoretical model proposed by the authors cannot be applied to the experiment. By the way, the STM topographic images are missing scale bars.

Our reply) We appreciate this comment. In response to the reviewer #1's first comment, we showed unambiguously the two pi phase shifts involved within a single soliton molecule and their chiral character (the existence of the phase shift only on one In chain of a wire). In order to make more clear the existence of two chiral solitons in the topographic image, we show below that the enlarged segments of a soliton molecule in fact correspond well to the images of the isolated chiral solitons with minor changes on the relative contrast of individual features. Of course, these distortions are expected from the soliton-soliton interaction to form a molecule. This figure was included in the supplements (Supplementary Fig. S11).

Figure caption) (a) High-pass filtered STM image for a soliton molecule shown in Fig. 1 and (b) enlarged images of the right and left part of the the soliton molecule. (c) Enlarged images of the isolated right- and left-chiral solitons shown in Fig. 1.

We added the scale bars in the corresponding STM topography.

Reviewer comment 2) It does not sound a common practice to interpret DFT result by randomly shifting the energies of the filled and empty states in the opposite directions (in order to make the experiment and theory look similar). I also do not agree that the authors' LDA result is in good agreement with the GW result even after energy shifting (shown in the response figure to last-round referee 1's comment 7, as a support for shifting the energy of the bands around). Therefore, it adds further doubts on the agreement between the authors' theory and experiment.

Our reply) We appreciate this comment. However, we would like to point out that we did not shift the filled and empty state randomly. What we did is to add a momentum-independent self-energy correction, whose magnitude is determined by the difference of the gap size between the LDA and GW calculation. There is no randomness and no arbitrary degree of freedom here. Correcting the difference between the LDA and GW approximations in gapped systems such as a CDW system has been extensively discussed [for example, Phys. Rev. B **85**, 195111 (2012)]. These studies tell that the difference comes from the exchange and correlation energy in the self energy. In the present system, what is more dominant is the exchange energy since the correlation part must be weak (we confirmed this by LDA+U calculations). The most widely used exchange energy correction is in fact to add a momentum-independent self-energy. This is the same as moving the filled and empty state band in opposite direction or increasing the gap size (see the above PRB paper). Except for fine details of the band dispersions affected by the exchange energy [the above reference and Phys. Rev. B **99**, 155107 (2019)], the overall difference of the band dispersions between LDA and GW approximations can be corrected in this way in many systems.

In the present case, the fine differences, which are uncorrected by this method is the band width of the filled band, see the red arrows in the figure below and the energy position of the Si bulk valence band (related to the band gap size of the Si bulk). The latter is not relevant for the present discussion of In wires. Note also that similar differences exist between the HSE and GW approximations and there exist Fermi level uncertainty in all these calculations in comparison to the experiment. We would like to empathize that what decides the topological property of the present system is the band dispersions around the X points indicated by green box below, where all approximations including our self-energy (band-gap) corrected LDA agree well. We elaborated our description on this issue within the manuscript.

Figure caption) Comparison of theoretical band structures. (a) Results of Heyd-Scuseria-Ernzerhof (HSE06) hybrid functional calculation [*Phys. Rev. B* 102, 121408(R) (2020)] with the present LDA calculation (blue circle). (b) GW band structure [*Phys. Rev. B* 99, 155107 (2019)] with the present LDA calculation (blue circle). (c) Enlarged band structures within the red boxes in (a) and (b). (d) Comparison between HSE06 and GW band structure.

One old issue: It seems that I still cannot find the exact locations where the Fig. 2 STS data were acquired. The black arrow in Fig. 1c points out the line where the data were taken, but the positions along that line seem to be not mentioned (or am I missing this information from the manuscript?).

Our reply) We appreciate this comment. Following this comment, we specified more accurately (the starting and end points added) where the STS data were taken in the manuscript. This is simply from one end of the image to the other.

REVIEWER COMMENTS

Reviewer #1 (Remarks to the Author):

I appreciate the authors for being patient and all the additional analysis in response to my questions. Overall, I think results are legitimate and the claims made in the manuscript are supported by them. The phase shift maps the author presented in the reply look very nice, visualizing the presence of the soliton molecule clearly. I think that equivalent figures should be presented in the main figure rather than putting in the supplementary material. So, I suggest that Fig. 1 (g) should be replaced with the phase shift map of the Fig. 1(f). That way, a new Fig. 1(g) would also be harmonized with Fig. 1(h), in which the authors have already highlighted the π phase shift as labels. Otherwise, I sign it off on a publication in Nature Communications.

Reviewer #2 (Remarks to the Author):

I thank the authors for their response to my questions, but I am still not fully convinced by their arguments.

1. If the distortion of the experimental data from the simulation close to the solitons is due to the soliton-soliton interaction as suggested by the authors, then in the manuscript the authors should point out this difference and comment why the interaction effect is not present in the simulation, otherwise by looking at such details in Fig. 1i and 1j one can easily get confused and question whether the extracted phase shift value is correct.

By the way, although the authors stated in their response that they have added scale bars in the revised manuscript, I still do not find them. Of course, one can derive the lateral feature size by counting the lattice sites, but I would think that it is good practice to add a scale bar in a STM image, and that's what a typical STM paper would do.

2. In the reference (PRB 85, 195111 (2012)) mentioned by the authors, a spatially uniform system TiSe₂ was studied and GW calculation was conducted. But in the manuscript the authors studied a system with distortions (the solitons and soliton molecules). The spatial inhomogeneity could in principle lead to momentum-dependent features. Therefore, I keep my doubt on whether a simple momentum-independent self-energy is enough to capture the physics here. In fact, I can see the discrepancies between the experiment and theory

(Fig. 2(a)(c)) even if a self-energy correction has been applied: both the empty states at ~ 300 mV and the occupied states at ~ -300 mV do not match. In the response figure I can also see differences in the empty states between LDA and GW (although for the filled states they agree well after exclusion of the Si states). To remove possible ambiguities, can the authors comment on these discrepancies? Is it possible to directly combine the self-energy correction in the band structure calculation for the soliton structure (e.g., by directly performing GW rather than LDA with energy shift)?

Point-by-point response to the comments of the reviewers:

Reviewer #1 (Remarks to the Author):

Reviewer comment) I appreciate the authors for being patient and all the additional analysis in response to my questions. Overall, I think results are legitimate and the claims made in the manuscript are supported by them. The phase shift maps the author presented in the reply look very nice, visualizing the presence of the soliton molecule clearly. I think that equivalent figures should be presented in the main figure rather than putting in the supplementary material. So, I suggest that Fig. 1 (g) should be replaced with the phase shift map of the Fig. 1(f). That way, a new Fig. 1(g) would also be harmonized with Fig. 1(h), in which the authors have already highlighted the p phase shift as labels. Otherwise, I sign it off on a publication in Nature Communications.

Our reply) We appreciate the continued effort of the reviewer with a helpful comment. Following the advice of the reviewer, we changed Fig. 1(g).

Reviewer #2 (Remarks to the Author):

Reviewer comment) I thank the authors for their response to my questions, but I am still not fully convinced by their arguments.

Our reply) We appreciate the continued effort of the reviewer. Overall, we understand that the reviewer is concerned about the discrepancy between the current DFT simulations (for both topography and for electronic states) and the experimental data. As we consistently argue, the current LDA-based DFT calculation is an approximation which has obvious limitation in its details in describing the In/Si(111) system even without a soliton molecule. However, this limitation has been well understood (without a soliton molecule) during the last decade by several groups and the method itself has been proved to be still effective in describing the CDW and the soliton formation of this system. In the present work, it is clear that this approximation tells properly about (i) the energetics of the molecule formation, (ii) the major structural characteristics with phase shifts, and (iii) the in-gap states (after the simplest self-energy correction). While one may be able to work with more advanced (much more time and cost consuming, at the same time) method, we are confident that it will only improve the details, delivering very little extra physics.

Reviewer comment 1) If the distortion of the experimental data from the simulation close to the solitons is due to the soliton-soliton interaction as suggested by the authors, then in the manuscript the authors should point out this difference and comment why the interaction effect is not present in the simulation, otherwise by looking at such details in Fig. 1i and 1j one can easily get confused and question whether the extracted phase shift value is correct.

By the way, although the authors stated in their response that they have added scale bars in the revised manuscript, I still do not find them. Of course, one can derive the lateral feature size by counting the lattice sites, but I would think that it is good practice to add a scale bar in a STM image, and that's what a typical STM paper would do.

Our reply) We appreciate the positive evaluation of our revision. Since the referee mentioned that he/she "can hardly see the AA chirality feature from the left region of the experimental data", we showed that phase shift is clearly present there and the (reserved-)AA feature in topography is experimentally very

clear as shown below. The distortion we mentioned in our previous reply is not the distortion between simulation and the experiments but the distortion in the experimental images from that of free isolated solitons (comparing the two sets of experimental data (b)/(c) shown below). These distortions do not prevent to identify the existence of two different types of solitons as shown below and as also confirmed by the quantitative measurement of the phase shift. Thus, we argue that the model we propose here, the existence of two solitons (chirally different), is fully validated.

The current level of agreement between simulated and experimental topography images is detailed again in the figure below. We believe that this level of agreement does not prevent to qualitatively prove the model proposed here. Indeed, this level of agreement is not much apart from that for the simulation/expt. of the pristine In chain structure without a soliton. Therefore, this is not an issue of including properly the soliton-soliton interaction in the simulation but an issue of the overall limitation of the current DFT approach for the In/Si(111) system itself, which we already discussed extensively. In the revised manuscript, however, considering the concern of the reviewer, we indicated the point where the DFT simulation deviates apparently from the experimental image (Fig. 1i and 1j).

We provided the length scale bars in the figure. We thought that the reviewer asked us to put the height scale bar since the length scale here is rather obvious.

Reviewer comment 2) In the reference (PRB 85, 195111 (2012)) mentioned by the authors, a spatially uniform system TiSe₂ was studied and GW calculation was conducted. But in the manuscript the authors studied a system with distortions (the solitons and soliton molecules). The spatial inhomogeneity could in principle lead to momentum-dependent features. Therefore, I keep my doubt on whether a simple momentum-independent self-energy is enough to capture the physics here. In fact, I can see the discrepancies between the experiment and theory (Fig. 2(a)(c)) even if a self-energy correction has been applied: both the empty states at ~ 300 mV and the occupied states at ~ -300 mV do not match. In the response figure I can also see differences in the empty states between LDA and GW (although for the filled states they agree well after exclusion of the Si states). To remove possible ambiguities, can the authors comment on these discrepancies? Is it possible to directly combine the self-energy correction in the band structure calculation for the soliton structure (e.g., by directly performing GW rather than LDA with energy shift)?

We are pleased to find that the reviewer now seems to agree that the energy shift is not arbitrarily done. We do not, however, agree that the present soliton structure requests a momentum-dependent energy shift since the energy levels of the soliton and soliton molecule are largely not dispersive. Moreover, the momentum-dependent energy shift may appear in the band structure but hardly in the total LDOS we are discussing. As mentioned by the reviewer, one can still find the discrepancy between the simulation and the experiment. We agree that the most prominent discrepancy is on the 300-meV peak. We explicitly mention this in the revised manuscript to make clear the limitation of the present simulation. This discrepancy, however, has been well known in the literature since it exists already for the pristine In chain structure (the 8x2 structure in the figure) and is not particularly related with the current soliton-molecule model. It has been well established that this state is responsible for the bonding between the outer part of the In chains with the substrate Si atoms. For example, we noted that the state around 300 meV (see S_3 in figure below) has a marginal impact on the CDW and solitons formation while the other two half-filled bands (S_1 and S_2) have dominating role [S. Cheon et al., *Science* 350(6257), 182-185 (2015)]. In fact, this issue for the DFT-LDA calculation and the S_3 band was already discussed in our previous reply. Since the most important spectral features of a soliton and a soliton molecule are their in-gap states, one can at least say that the current simulation captures the emergence of the soliton molecule's in-gap state above the valence band edge only with a marginal energy shift (now indicated explicitly in the figure). We mentioned this discrepancy briefly in the revised manuscript.

While the reviewer mentioned performing a GW calculation, it is a formidable task, practically impossible, considering the huge supercell used for the current structure model.

S. Cheon et al., *Science* 350 (6257), 182-185.

(Left: the DFT-LDA calculation of the band dispersion of In/Si(111) for the undistorted and distorted structures. Center: the essential tight-binding model of the same system, which describes the CDW and soliton formation. Right: the current DFT-LDA calculation for the CDW structure).

REVIEWERS' COMMENTS

Reviewer #2 (Remarks to the Author):

I thank the authors for their patience in answering my previous questions. Now my remaining concerns are cleared and I therefore support publication of the manuscript in Nature communications.